# Spatio-temporal prediction model of out-of-hospital cardiac arrest: Designation of medical priorities and estimation of human resources requirement

**Angelo Auricchio[1,2]\*, Stefano Peluso[3,4], Maria Luce Caputo[1,5], Jost Reinhold[3], Claudio Benvenuti[2], Roman Burkart[2], Roberto Cianella[6], Catherine Klersy[7], Enrico Baldi [1,5,8], Antonietta Mira[3,9]**

**1** Fondazione TicinoCuore, Breganzona, Switzerland, **2** Division of Cardiology, Cardiocentro Ticino, Lugano, Switzerland, **3** Data Science Lab, Institute of Computational Science, Università della Svizzera italiana, Lugano, Switzerland, **4** Department of Statistical Sciences, Università Cattolica del Sacro Cuore, Milan, Italy, **5** Department of Molecular Medicine, Section of Cardiology, University of Pavia, Pavia, Italy, **6** Federazione Cantonale Ticinese Servizi Autoambulanze, Lugano, Switzerland, **7** Unit of Clinical Epidemiology & Biometry, IRCCS Fondazione Policlinico san Matteo, Pavia, Italy, **8** Cardiac Intensive Care Unit, Arrhythmia and Electrophysiology and Experimental Cardiology, Fondazione IRCCS Policlinico San Matteo, Pavia, Italy, **9** Department of Science and High Technology, University of Insubria, Como, Italy

\* angelo.auricchio@cardiocentro.org

**Data Availability Statement:** All relevant data are within the manuscript and its Supporting Information files.

## Abstract

### Aims

To determine the out-of-hospital cardiac arrest (OHCA) rates and occurrences at municipality level through a novel statistical model accounting for temporal and spatial heterogeneity, space-time interactions and demographic features. We also aimed to predict OHCAs rates and number at municipality level for the upcoming years estimating the related resources requirement.

### Methods

All the consecutive OHCAs of presumed cardiac origin occurred from 2005 until 2018 in Canton Ticino region were included. We implemented an Integrated Nested Laplace Approximation statistical method for estimation and prediction of municipality OHCA rates, number of events and related uncertainties, using age and sex municipality compositions. Comparisons between predicted and real OHCA maps validated our model, whilst comparisons between estimated OHCA rates in different yeas and municipalities identified significantly different OHCA rates over space and time. Longer-time predicted OHCA maps provided Bayesian predictions of OHCA coverages in varying stressful conditions.

### Results

2344 OHCAs were analyzed. OHCA incidence either progressively reduced or continuously increased over time in 6.8% of municipalities despite an overall stable spatio-temporal distribution of OHCAs. The predicted number of OHCAs accounts for 89% (2017) and 90%

**Funding:** The study was supported by a grant of the Swiss Heart Foundation (Bern, Switzerland) "Identification of high-risk areas of out-of-hospital cardiac arrest in Switzerland." Partial support was also obtained by Fondazione Fratelli Agostino Enrico Rocca.

**Competing interests:** AA is a consultant to Boston Scientific, Backbeat, Biosense Webster, Cairdac, Corvia, Microport CRM, Philips, Radcliffe Publisher. He received speaker fee from Boston Scientific, Medtronic, and Microport. He participates in clinical trials sponsored by Boston Scientific, Medtronic, Philips. He has intellectual properties with Boston Scientific, Biosense Webster, and Microport CRM. All other authors have no conflict of interest to disclose. This does not alter our adherence to PLOS ONE policies on sharing data and materials.

(2018) of the yearly variability of observed OHCAs with prediction error ≤1OHCA for each year in most municipalities. An increase in OHCAs number with a decline in the Automatic External Defibrillator availability per OHCA at region was estimated.

## Conclusions

Our method enables prediction of OHCA risk at municipality level with high accuracy, providing a novel approach to estimate resource allocation and anticipate gaps in demand in upcoming years.

## Introduction

Out-of-hospital cardiac arrest (OHCA) is a major public health problem in western countries, with the North American incidence of emergency medical system (EMS)-assisted cardiac arrest estimated to vary between 71.8 to 159 per 100,000 people [1]. OHCA incidence, rate of return-to-spontaneous circulation (ROSC), and survival to discharge of EMS-treated differ significantly across geographic regions [2,3]. However, regional variations are observed across areas as small as neighbourhoods [4,5] and have been attributed to population density [6,7], socio-demographic characteristics of the population [8,9], and to education level [10].

Designation of medical priorities based on disease distribution is important to identify immediate precipitants of the disease itself, to develop preventative strategies, and most importantly to optimize resource planning. As shown by Chocron et al. [11], sufficient allocation of basic life support and advanced life support ambulances per served population was associated with higher ROSC rate and survival. Another application of spatial OHCA distribution is mathematical optimization of automated external defibrillator (AED) placements [12–15]. All studies evaluating national, regional, and neighbourhood distribution of OHCA have exclusively focused on the spatial domain without accounting for the temporal dimension. However, spatio-temporal heterogeneity of OHCA may exist, although this has yet to be studied. If it exists, it may have major implications on resource planning and its optimization. For example, in the event of significant spatio-temporal heterogeneity in the OHCA distribution, coverage by AED may become inadequate or suboptimal over time. Iterative optimization models of AED placement are currently not available, and a key question is the length of the iteration cycle (every year, every 5–10 years), considering that AED relocation or a new AED installation is a costly effort [16]. Finally, assessment of spatio-temporal heterogeneity may provide an opportunity to predict community or neighbourhood OHCA distributions in the upcoming years, a wishful effort not yet investigated. Indeed, in the modern era of activation of mobile lay rescuers (i.e. first responders), mobile phone positioning systems are linked to the emergency medical dispatch centre, and then prediction of OHCA distribution could significantly reduce response time to alert.

In the present study, through a novel statistical model accounting for temporal and spatial heterogeneity, space-time interactions and demographic features, we aim to determine the OHCA rates and occurrences at municipality level in the area of Canton Ticino, Switzerland, from 2005 to 2018. Then we predict the respective OHCAs rates and number of instances for the upcoming years, under different assumptions of changes in population profile, with the purpose of estimating the related resource requirements.

## Methods

### Population

All individuals being older than 1 year who suffered an OHCA in the Swiss Canton Ticino region from 1 January 2005 until 31 December 2018 have been included in the study. The coverage of OHCA cases was complete because the EMS system is activated for all emergencies involving a cardiac arrest. The definition excludes cases with obvious late signs of death (e.g. rigor mortis) for which resuscitative efforts are not initiated or when a do-not-resuscitate order is in place. OHCA was defined as cessation of cardiac mechanical activity, confirmed by the absence of signs of circulation, occurring outside of a hospital setting. A bystander was defined as an individual who witnessed the collapse or who found the person unresponsive and activated the EMS system.

### Registry

The Ticino Registry of Cardiac Arrest (TIRECA) is a web-based, prospectively designed registry, and has the goal to monitor OHCA in the Swiss Canton Ticino as well as to identify potential areas for improvement in cardiac and emergency care. It has been described previously [17]. In short, the registry was established on January 1st, 2002; however, consecutive and audited data have been entered starting on January 1st, 2005. It contains a record of every individual who presented a cardiac arrest of any etiology, and includes patient's demographic data, comprehensive EMS-related data, detailed bystander and first responder activity including the use of AED or Public Access Defibrillator (PAD) as well as pre- and in-hospital treatment and outcome. Prior to April 2009, OHCA events were manually geolocated based on the address provided by the ambulance; all subsequent OHCAs were automatically geolocated. Data are collected and stored following Good Clinical Practice Guidelines and the relevant legislation governing the use of patient data. The investigation complied with the Declaration of Helsinki's principles for physicians engaged in biomedical research involving human subjects and was approved by the appropriate ethics committee.

### Geographical and municipalities data

As of December 31st, 2018 Swiss Canton Ticino presents 117 municipalities. Gender composition and age distribution have been retrieved from the Land Register of Canton Ticino (https://www4.ti.ch/di/dg/sr/home/). The geographical configuration of municipalities has changed over the studied years through municipality mergers and splits, we have therefore reconstructed the time series so that the data are consistent with the most recent available Canton Ticino map in 2017.

### Statistical analysis

The methodology of this study is consistent with the STROBE (Strengthening The Reporting of Observational Studies in Epidemiology) checklist for observational studies. We imported the data from Microsoft Access into Stata 16.0 (StataCorp, College Station, TX, USA) and cleaned them with back and forth checks with the registry data manager, before undergoing analysis. We express continuous data as mean and standard deviation (SD) or median and 25% to 75% percentiles. We express categorical data as frequencies and percentages. We compare data among meaningful groups using the Kruskall-Wallis test and the Fisher exact or the Chi-squared test, respectively. All tests are two sided and significance was set at 5%. The Bonferroni correction was applied for post-hoc comparisons.

To determine the yearly overall and at municipality level incidence of OHCA, i.e. number of new cases per unit of person-time at risk on every year, we divide the yearly number of OHCAs by the local population at risk during the same period. We test spatial heterogeneity with a Tango's cluster detection test [18] and the presence of temporal trends with a chi-squared test for trend in proportions.

We use the statistical software R [19] for data analysis. In particular, we use the implementation of Integrated Nested Laplace Approximation (INLA), packages *maptools* and *raster* for reading the shape file of Canton Ticino [20,21], and *spatstat*, *rgdal* and *spdep* for general spatial statistical tools [22–24]. We adopt INLA to estimate OHCA incidences with related uncertainty, and to predict OHCA incidences (and related uncertainty) in future years. The methodology for the development of the INLA-based statistical model for the spatio-temporal analysis of OHCA data has been recently reported [25,26]. We split our sample from 2005 to 2018 in a training (2005–2016) and a validation sample (2017–2018) over which our predictions are tested. To verify that the goodness of prediction does not deteriorate in longer forecast periods, we perform a sensitivity analysis of our results by predicting years 2017 and 2018 by keeping fixed the training sample from 2005 to 2016. We also perform forecasts based on an adaptive sample size: OHCAs in years 2005–2016 are used to predict OHCAs in 2017, whilst OHCAs in years 2005–2017 are used to predict 2018. We finally predict longer-term OHCAs in 2023 and the number of needed AED, under three different increasingly severe demographic scenarios: a) population as in 2018; b) population growth following the historical trend; c) population on average older than 5%, relative to 2018. The coefficient of determination, R2, for a linear regression model between true and estimated/predicted OHCAs is adopted as proportion of variability in OHCAs explained by the estimated number of OHCAs, and therefore a higher R2 corresponds to a better performing statistical model and to a better explanation of space-time OHCA variability across municipalities.

## Results

### Population

Over the entire study period, 3221 OHCAs have been registered. OHCAs of non-cardiac origin (877, 27.2%) were excluded, and the remaining 2344 OHCAs of presumed cardiac origin represents the study population.

Patients and resuscitation characteristics are shown in Table 1. Resuscitation was attempted in 1418 cases, and 467 fulfilled the Utstein comparator criteria. Out-of-hospital cardiac arrest occurred prevalently at home (66.2%), in men (71.2%) of a mean age of 71±14 years. There was a slight but significantly increase in the mean population age over the years (p = 0.03), whilst there were no statistically significant differences for gender, OHCA location, percentage of unwitnessed and median EMS arrival time (Table 2). A statistically significant fluctuation over the years regarding the presenting rhythm was revealed (p = 0.03), although without a significant trend. As shown in Table 2, there is a statistically significant increase in the percentage of bystander CPR (p<0.001), with a reduction of the time from call to CPR (p<0.001), and in the percentage of shock delivered before EMS arrival (p<0.001). Since 2005 the OHCA survival rate and the rate for those witnessed OHCA of cardiac origin which presented a shockable rhythm, progressively increased reaching, in 2018, 32% and 60%, respectively (Table 2).

### OHCA rates at municipality level

In order to avoid bias regarding incidence and prediction of OHCA, cases with missing geolocation coordinates (98 cases, 4.2%) were excluded. There was a negligible annual fluctuation in the absolute number of OHCA over the entire region (p = 0.42). Fig 1 shows the cumulative (year

**Table 1. Demographic characteristics of the 2344 out-of-hospital cardiac arrest of presumed cardiac origin occurring in Canton Ticino from 2005 until 2018.**

| Variable | Value |
|---|---|
| **Age (yrs)** | 71 ± 14 |
| **Men (n, %)** | 1668 (71.2%) |
| **Home location OHCA (n, %)** | 1552 (66.2%) |
| **Time from call to EMS arrival, mins (median, IQR)** | 10 [8–14] |
| **Unwitnessed OHCA (n, %)** | 915 (39%) |
| **Bystander CPR (n, %)** | 1418 (60.5%) |
| **Time from call to CPR, mins (median, IQR)** | 6 [3–10] |
| **First rhythm shockable (n, %)** | 815 (35.9%) |
| **Shock before EMS arrival (n, %)** | 140 (6%) |
| **Sustained ROSC (n, %)** | 810 (34.6%) |
| **Time from event to ROSC, mins (median, IQR)** | 26 [19–36] |
| **Survival at hospital discharge (n, %)** | 477 (20.4%) |

2005–2018) spatial distribution of OHCA due to cardiac etiology with the absolute OHCA number for each municipality. Both absolute cumulative (year 2005–2018) number of OHCAs, and OHCA incidence were heterogeneously distributed over space (both p<0.001) (S1 Fig). As shown in Fig 1 and in the S1 Fig, there was a spatial heterogeneity in the absolute number of events and OHCA incidence for males (p<0.001 for absolute events, p = 0.003 for incidences) and for absolute number of OHCAs for females (p = 0.002), but not for female incidences (p = 0.2).

## Prediction of OHCA rate at municipality level

The training set consisted of OHCAs occurred up to 2016 whilst the prediction was confined to 2017 (one-year prediction) and to 2018 (two-year prediction). The predicted absolute

**Table 2. Trend of bystander CPR, time from call to CPR, shock before EMS arrival, sustained ROSC, survival at hospital discharge in the whole population and survival at hospital discharge in the Utstein group (i.e. cardiac, bystander witnessed, shockable) over the years.**

| Year | Bystander CPR (n, %) | Time from call to CPR, Mins (median, IQR) | Shock before EMS arrival (n, %) | Sustained ROSC (%) | Survival at hospital discharge (%) | Survival at hospital discharge: Utstein group (%) |
|---|---|---|---|---|---|---|
| **2005** | 47 (29.4) | 9 [7–12] | 0 (0) | 46 (28.8) | 12 (7.5) | 7 (20) |
| **2006** | 84 (50) | 8 [2–12] | 0 (0) | 52 (31) | 28 (16.7) | 13 (35.1) |
| **2007** | 66 (44.6) | 8 [4–12] | 0 (0) | 41 (27.7) | 22 (14.9) | 10 (47.6) |
| **2008** | 86 (54.8) | 6 [2–10] | 0 (0) | 61 (38.9) | 33 (21) | 16 (50) |
| **2009** | 94 (51.9) | 8 [4–12] | 9 (5) | 62 (34.2) | 34 (18.8) | 22 (51.2) |
| **2010** | 99 (55.9) | 7 [3–11] | 14 (8) | 52 (29.4) | 24 (13.6) | 16 (57.1) |
| **2011** | 100 (58.8) | 7 [2–11] | 10 (6) | 52 (30.6) | 27 (15.9) | 14 (50) |
| **2012** | 103 (63.2) | 5 [3–9] | 20 (12) | 56 (34.4) | 32 (19.6) | 18 (56.2) |
| **2013** | 125 (66.8) | 5 [2–8] | 16 (9) | 67 (35.8) | 42 (22.5) | 22 (57.9) |
| **2014** | 113 (72.9) | 5 [3–9] | 20 (13) | 71 (45.8) | 51 (32.9) | 20 (69) |
| **2015** | 138 (83.1) | 5 [3–9] | 17 (10) | 53 (31.9) | 29 (17.5) | 7 (43.8) |
| **2016** | 123 (75.5) | 5 [4–9] | 10 (6) | 54 (33.1) | 42 (25.8) | 18 (66.7) |
| **2017** | 118 (69) | 5 [4–10] | 8 (5) | 63 (36.8) | 44 (25.8) | 15 (57.7) |
| **2018** | 122 (68.5) | 5 [3–10] | 16 (9) | 80 (44.9) | 57 (32) | 22 (71) |
| **P value** | P<0.001 | P<0.001 | P<0.001 | P = 0.006 | P<0.001 | P = 0.004 |

## A: All     B: Male     C: Female

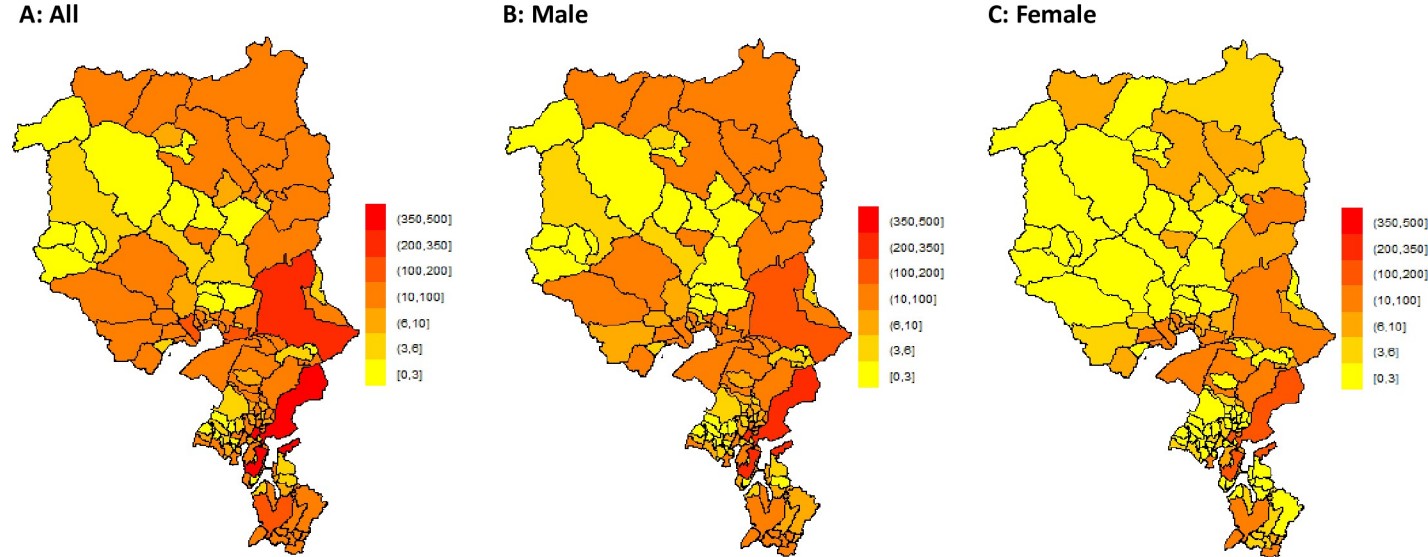

**Fig 1. Spatial distribution (at municipality level) of OHCAs expressed as absolute number due to cardiac etiology and according to gender of OHCA victim.**

number of OHCAs accounts for 89% (year 2017) and 90% (for year 2018) of the yearly variability of observed OHCAs (Fig 2, panel A & B). The predicted number of OHCAs for the year

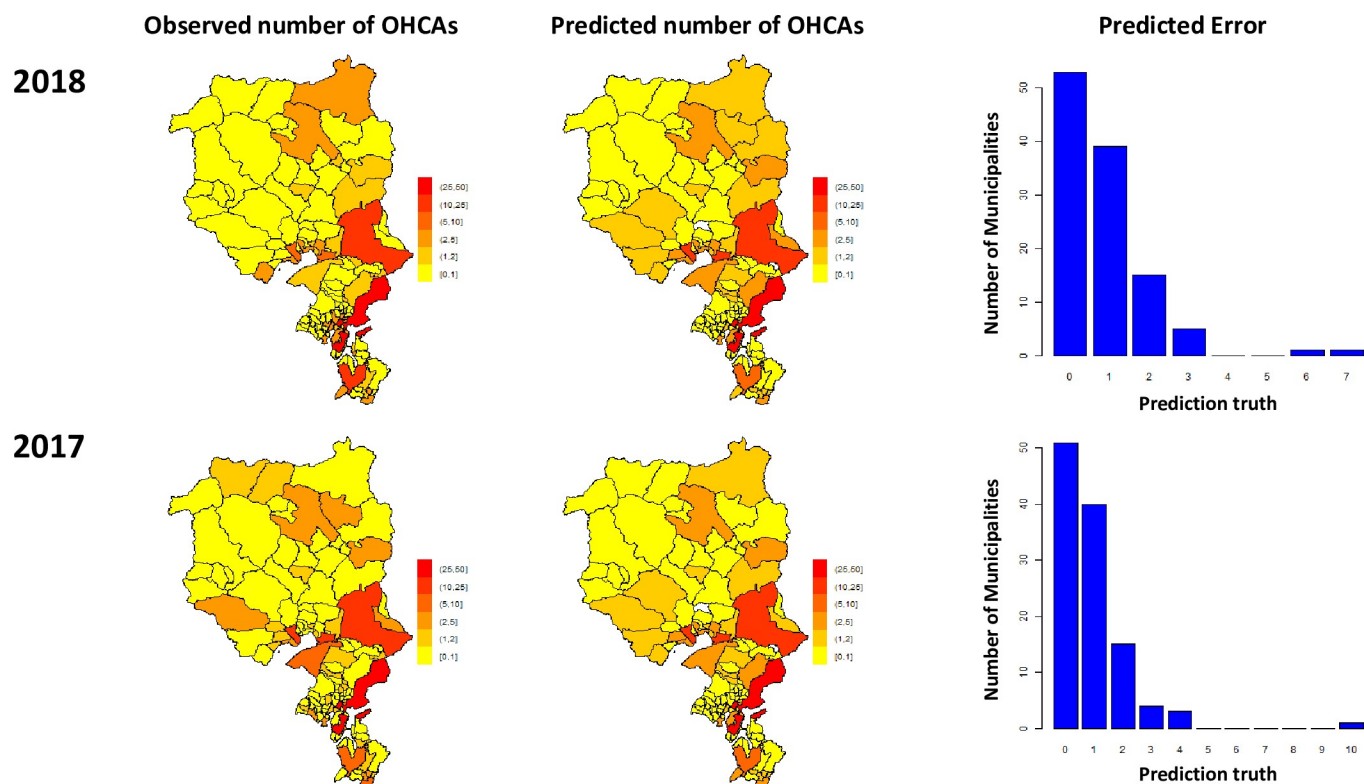

**Fig 2. Distribution of the events at the municipality level as observed in year 2017 and 2018 (panel A), and the predicted OHCA number of events as estimated by our method (panel B).** Panel C shows the distribution of the number of municipalities according to the predicted error (|observed-predicted|). In the majority of cases, there was zero difference in the absolute number of OHCA between the predicted and observed number of OHCAs.

2018 showed no loss of prediction accuracy from one-year to two-year forecasts. Fig 2 shows the prediction error distribution of OHCAs at municipality level; in the vast majority of municipalities the prediction error was equal to or less than one OHCA for each of the analysed year. In 8 of 117 municipalities (6.8%), there was a sign of temporal trend in the overall OHCA incidence; in 4 of 117 (3.4%) municipalities, the trend of OHCA incidence was significantly (p<0.05) downwards whereas in 4 of 117 (3.4%) municipalities, the trend was significantly upward (p<0.05). The clinical characteristics of OHCA patients occurring at each municipality with unchanged temporal trend, downward trend and upward trend are shown in S1 Table. No significant differences were shown between the three groups, other than for the time from call to EMS arrival, that was significantly longer than in the unchanged group (13 vs. 10 minutes, p<0.001 after Bonferroni correction).

The ability to predict OHCA incidence at the municipality level, together with its credible interval, allows for direct comparisons between municipalities over time, as exemplified in Fig 3 for three different head to head comparisons between municipalities; overall, our data elicit several possible patterns over space and time, expression of spatio-temporal heterogeneity in OHCA incidences across municipalities.

## Estimation of resources

Each EMS area includes on average 20 municipalities. Three different scenarios were designed in order to model the needed AED resources in each region. A first conservative approach considered the same population with unaltered gender distribution and age strata as in 2018. A second scenario was designed to estimate the population growth following the historical trend seen since year 2005 and finally, a third scenario considered a population older than the one observed in 2018 by 5%. Irrespective of the considered scenario, in the 4 out of 5 EMS areas an increase in the absolute number of OHCAs with a corresponding decline in the AEDs availability per estimated OHCA at EMS region was observed (Fig 4). Only in one EMS area, a disproportionate coverage of AED compared to expected number of OHCAs was highlighted.

## Discussion

To the best of our knowledge, we created the first risk map that estimates spatio-temporal OHCA incidence at municipality level. Although there was an overall stable spatio-temporal distribution of OHCA, we noted the existence of municipalities where OHCA incidence either progressively reduced or continuously increased over time. Another novel aspect of our work is represented by its ability to predict future OHCA events. Indeed, the adopted statistical model enables estimation of OHCA incidence with good accuracy at municipality level for the upcoming few years. An immediate application of our approach may be found in the evaluation of AED coverage which allows prediction of whether currently allocated resources may become insufficient or redundant over time with population changes, thus resulting in corrective actions.

The adopted statistical methodology is the Integrated Nested Laplace Approximation that, relative to the past literature [25,26], has now found a novel application in the evaluation of OHCA distribution at municipality level located in rural and urban areas spread over a large territory with valley, mountains, and rivers. Consistent with the results of Lin et al [26], we noticed that INLA is a highly efficient method and provides valid answers within few minutes [27], against huge computational efforts of more traditional spatio-temporal estimation techniques. Furthermore, INLA is a Bayesian approach, and therefore can quantify the uncertainty in the estimation of OHCAs and related incidences. The first advantage of the uncertainty quantification is that one can associate to each OHCA incidence (in a specific municipality

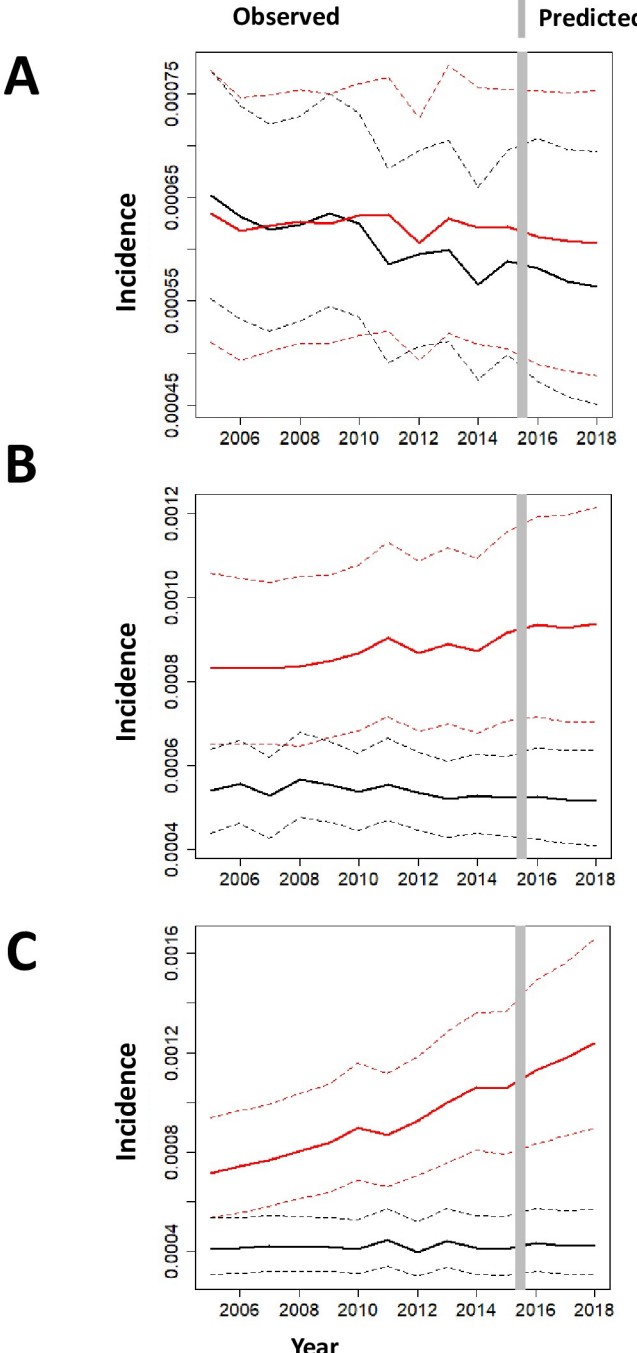

**Fig 3.** Each panel (A-C) shows the estimated OHCA incidences from 2005 to 2016 in a given municipality (red) with corresponding 95% credible intervals in dashed lines, and it compares to another municipality (black). In panel A, the 2 municipalities had similar incidence over time; in panel B and C, there is a progressive increase in the number of OHCA in one municipality compared to the other one. Panel A shows two municipalities where no significant difference in the OHCA incidence all over the sample space, and no difference is predicted in the upcoming years. Panel B shows the comparison between two municipalities in which OHCA incidence is significantly different over the whole sample. The distance between the two municipalities increased over time with an unchanged trend predicted over the validation sample (year 2017–2018). The increasing difference between the two OHCA incidences seems to be related to a deterioration of OHCA incidence in one municipality whilst incidence in the other municipality appears quite stable over time. Finally, panel C shows two municipalities with mixed results. Indeed from 2005 to 2008, uncertainties in the estimates of OHCA incidence do not allow us to distinguish the two incidence rates in a statistically significant way, but from 2009 on the two incidences become significantly different, and with significance

increasing over time, including the validation sample up to 2018. The difference seems to be attributed to an increase in OHCA incidence in one municipality together with a stabilization and then lower uncertainty on the OHCA incidence of the other municipality.

and year) a degree of belief in the estimated value. Of course, municipalities and years for which OHCA incidences are estimated from more OHCA events and higher populations have less degree of uncertainty, and therefore present a more reliable estimate of OHCA incidence. The second advantage is that it permits the identification of municipalities showing different incidences. Our results are easily interpretable, and suitable to prediction of future OHCAs.

Our study significantly expands knowledge in developing OHCA risk maps, since it explicitly includes the time dimension, so far neglected in most previous studies. Lin et al. [26] evaluated spatial accessibility of OHCAs through INLA, and developed a priority ranking to identify the greatest gaps between demand and supply of allocations, but only 3 years of historical data were available, and the time dimension was not considered. Also, Chan et al. [12,13], using a different approach based on logistic regression with kriging, identified spatial (but not temporal) heterogeneity in emergency medical resources between rural and urban areas. Finally, Sun et al. [14] identified and ranked business locations by spatio-temporal OHCA coverage. They also studied the temporal stability of the ranking, but with a method that, compared with ours, requires larger computational efforts, is more time-consuming and cannot consider geographic constraints (e.g. mountains, lakes, rivers, etc.). Their study was also limited to a single urban area (Toronto), in contrast to the region we study, where both rural and urban areas are included.

**Fig 4. Expected changes in the number of OHCAs according to three different scenarios and respective coverage by AED for each regional EMS.**

Our data show that in some municipalities the time dimension can be critical because OHCA incidence either progressively increased or decreased over time. These municipalities represented only a small proportion (about 7%) of all municipalities, still it suggests that time can be important and is not simply an a priori effect. Confirmation of these findings in other geographies may provide a unique opportunity for primary prevention interventions and possibly OHCA reductions.

OHCA is a time-sensitive disease, thus evaluation of its spatio-temporal distribution is of paramount importance for appropriate coverage by emergency medical resources [28,29], AEDs [30], and (possibly even more) by first responders [31,32]. Therefore, prediction of OHCA risk at regional or municipality level may advance resuscitation science, through a unique novel approach to estimate resource allocation and anticipate gaps in demand for upcoming years in defined areas. In spite of a small annual event rate, in about 80% of municipalities a very low OHCA prediction error at municipality level was found (less than or equal to one OHCA). Importantly, we noticed no loss of prediction accuracy from one-year to two-year forecasts indicating the robustness of the method, though we recognize the need for further validation in other geographies.

To improve the likelihood of AED usage in cardiac arrests, a novel data-driven framework for public AED placement, integrating OHCA risk estimation with AED location optimization, has been developed. The mathematical optimization models of AED placement use historical OHCA series and, considers cluster detection, i.e. a set of events that are spatially closely related. Recent research has indicated that optimization models for AEDs placement are superior to population models, thus adoption of such models could be considered by communities when selecting areas for AED deployment [12–14,26]. Compared to other models, flexible AED location models increase overall OHCA coverage, and decreases the distance to nearby AEDs, even in rural areas, while saving significant financial resources [15].

Regional differences in incidence might be partly related to differences in the completeness of case ascertainment and potential for undetected cases. However, each site had or implemented approaches to ascertain cardiac arrests from all EMS agencies within their geographic area. This prospective approach, in conjunction with statistical methods to account for missing cases, provides the most robust resource to date for determining the public health magnitude of cardiac arrest. Thus, the observed differences in incidence most likely reflect differences in the underlying risk of OHCA, as well as the local approach to organized emergency response and post-resuscitation care in the hospital. Another aspect is the exclusion of patients with missing geolocation coordinates: this could potentially bias OHCA incidence estimation and prediction; however, this group was extremely small (5% of the total population), thus unlikely to affect our prediction model.

## Conclusions

Estimation of spatio-temporal OHCA incidence at municipality level is feasible. On the background of an overall stable spatio-temporal OHCA incidence, there were several municipalities in which OHCA steadily increased or decreased over time, a finding which needs to be further validated in other geographies. This observation may provide a unique opportunity for primary prevention intervention in selected municipalities. Finally, our method enables prediction of OHCA risk at regional or municipality level with high accuracy, thus providing a novel approach to estimate resource allocation and anticipate gaps in demand in upcoming years in a certain area.

## Supporting information

**S1 Fig. Spatial distribution (at municipality level) of OHCAs expressed as incidence due to cardiac etiology for the whole population and according to gender of OHCA victim.** Incidence was calculated according to the inhabitants of each municipality and according to the inhabitants of each gender for each municipality for the gender-related analysis.
(PDF)

**S1 Table. Demographic characteristics of the out-of-hospital cardiac arrest by dividing municipalities according to the trend in OHCA incidence observed from 2005 until 2018.**
(DOCX)

## Acknowledgments

We would like to thank all the researchers and personnel involved in the Ticino Registry of Cardiac Arrest (TIRECA).

## Author Contributions

**Conceptualization:** Angelo Auricchio, Stefano Peluso, Antonietta Mira.

**Data curation:** Claudio Benvenuti, Roman Burkart, Roberto Cianella.

**Formal analysis:** Stefano Peluso, Jost Reinhold, Catherine Klersy, Antonietta Mira.

**Funding acquisition:** Angelo Auricchio.

**Investigation:** Claudio Benvenuti, Roman Burkart, Roberto Cianella.

**Supervision:** Angelo Auricchio, Antonietta Mira.

**Writing – original draft:** Angelo Auricchio, Stefano Peluso, Maria Luce Caputo.

**Writing – review & editing:** Enrico Baldi, Antonietta Mira.

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
