## [Decision Letter · Decision Letter 0]

10 Aug 2020

Spatio-temporal prediction model of out-of-hospital cardiac arrest: designation of medical priorities and estimation of human resources requirement.

PONE-D-20-10664

Dear Dr. Auricchio,

We’re pleased to inform you that your manuscript has been judged scientifically suitable for publication and will be formally accepted for publication once it meets all outstanding technical requirements.

Kind regards,

Andrea Ballotta

Academic Editor

PLOS ONE

1. Thank you for stating the following in the Competing Interests section:

[AA is a consultant to Boston Scientific, Backbeat, Biosense Webster, Cairdac, Corvia, Microport CRM, Philips, Radcliffe Publisher. He received speaker fee from Boston Scientific, Medtronic, and Microport. He participates in clinical trials sponsored by Boston Scientific, Medtronic, Philips. He has intellectual properties with Boston Scientific, Biosense Webster, and Microport CRM. All other authors have no conflict of interest to disclose.].

Please respond by return email with your amended Competing Interests Statement and we will change the online submission form on your behalf.

Additional Editor Comments (optional):

Thank you for contribution. On the basis of the reviewer's comments i've deemed the paper suitable for publication. Congratulations.

Reviewers' comments:

Reviewer's Responses to Questions

**Comments to the Author**

1. Is the manuscript technically sound, and do the data support the conclusions?

Reviewer #1: Yes

2. Has the statistical analysis been performed appropriately and rigorously? 

Reviewer #1: Yes

3. Have the authors made all data underlying the findings in their manuscript fully available?

Reviewer #1: Yes

4. Is the manuscript presented in an intelligible fashion and written in standard English?

Reviewer #1: Yes

5. Review Comments to the Author

Reviewer #1: Thanks for your paper.

That is an impressive job and a good tool for prediction of the EMS and AED need on the OHCA.

It would be a good idea to implement it on a higher scale to see what is the result

6. PLOS authors have the option to publish the peer review history of their article (what does this mean?). If published, this will include your full peer review and any attached files.

Reviewer #1: No

---

## [Editor Report · Acceptance letter]

13 Aug 2020

PONE-D-20-10664 

Spatio-temporal prediction model of out-of-hospital cardiac arrest: designation of medical priorities and estimation of human resources requirement. 

Dear Dr. Auricchio:

I'm pleased to inform you that your manuscript has been deemed suitable for publication in PLOS ONE. Congratulations! Your manuscript is now with our production department. 

Kind regards, 

on behalf of

Dr. Andrea Ballotta 

Academic Editor

PLOS ONE